# Ligand-Induced GPR110 Activation Facilitates Axon Growth after Injury

**DOI:** 10.3390/ijms22073386

**Published:** 2021-03-25

**Authors:** Heungsun Kwon, Karl Kevala, Hu Xin, Samarjit Patnaik, Juan Marugan, Hee-Yong Kim

**Affiliations:** 1Laboratory of Molecular Signaling, NIAAA, National Institutes of Health, 5625 Fishers Lane Room 3S-02, Rockville, MD 20892, USA; heung-sun.kwon@nih.gov (H.K.); karl.kevala@nih.gov (K.K.); 2Division of Pre-Clinical Innovation, NCATS, National Institutes of Health, Rockville, MD 20852, USA; xin.hu@nih.gov (H.X.); patnaiks@mail.nih.gov (S.P.); maruganj@mail.nih.gov (J.M.)

**Keywords:** GPR110, synaptamide, cAMP/PKA, axon, optic nerve, microfluidic culture platform, retinal explant culture

## Abstract

Recovery from axonal injury is extremely difficult, especially for adult neurons. Here, we demonstrate that the activation of G-protein coupled receptor 110 (GPR110, ADGRF1) is a mechanism to stimulate axon growth after injury. *N*-docosahexaenoylethanolamine (synaptamide), an endogenous ligand of GPR110 that promotes neurite outgrowth and synaptogenesis in developing neurons, and a synthetic GPR110 ligand stimulated neurite growth in axotomized cortical neurons and in retinal explant cultures. Intravitreal injection of GPR110 ligands following optic nerve crush injury promoted axon extension in adult wild-type, but not in *gpr110* knockout, mice. In vitro axotomy or in vivo optic nerve injury rapidly induced the neuronal expression of *gpr110*. Activating the developmental mechanism of neurite outgrowth by specifically targeting GPR110 that is upregulated upon injury may provide a novel strategy for stimulating axon growth after nerve injury in adults.

## 1. Introduction

Despite extensive efforts to develop repair strategies, injured CNS nerves remain difficult to regenerate in adulthood [1], largely due to inhibited intrinsic axon growth capacity [2], the formation of a myelin-associated growth-inhibitory environment and scarring around the lesion site [3], and increased neuronal cell death [4]. However, axon regeneration has been demonstrated through manipulating genes or signaling pathways to stimulate axon growth potential and to promote neuronal survival [5,6]. The induction of cyclic adenosine monophosphate (cAMP) signaling has also emerged as a promising strategy for stimulating axon growth and nerve cell migration across inhibitory substrates [7,8,9]. Previously, we have demonstrated that *N*-docosahexaenoylethanolamine (synaptamide), an endogenous metabolite of docosahexaenoic acid (DHA, 22:6n-3), potently promotes neurite outgrowth and synaptogenesis by binding to the G-protein coupled receptor 110 (GPR110, ADGRF1) and increasing cAMP in developing neurons [10,11]. GPR110 is an adhesion GPCR, and its expression in the brain is highest during development but diminishes in adulthood [10,11]. In the present study, we tested whether this endogenous mechanism for developmental neurite outgrowth is applicable to axon growth after injury. We found that GPR110 activation by an endogenous or synthetic ligand can stimulate axon growth after injury and after crush-induced optic nerve injury in adult mice.

## 2. Results

### 2.1. Axon Growth Is Stimulated by Somal or Axonal Treatment with Synaptamide

We have previously demonstrated that synaptamide stimulated neurite growth in cortical cultures with an IC_50_ value in the low nM range [10]. We tested whether synaptamide-induced axon growth is due to somal or axonal stimulation of GPR110 using cortical neurons cultured in a microfluidic culture platform, where two chambers are separated by 500 µm of thin grooves (Figure 1A) [12]. We observed stimulated axon growth after either somal or axonal application of synaptamide (Figure 1B,C), as expected by the *gpr110* expression in both soma and distal axons (Figure 1D). The presence of GPR110 in the distal axons was further indicated by the endocytic puncta that were visualized by bodipy–synaptamide, a fluorescent GPR110 ligand [10], and were colocalized with clathrin (Figure 1E).

### 2.2. Synaptamide Stimulates Axon Regeneration In Vitro through GPR110/cAMP/PKA-Dependent Signaling

To examine axon regeneration, we first established an in vitro axon injury model using the microfluidic culture platform. Axons from primary cortical neurons seeded in one chamber were grown sufficiently through these grooves into the other chamber before axotomy was performed.

Addition of 10 nM synaptamide to either the somal (Figure 2A,B) or axonal compartment (Figure 2C,D) after axotomy produced a 3.5- or 2.2-fold increase in axon regrowth, respectively, compared to the vehicle control at 7 days after treatment. No such effect was observed when axotomized axons were treated with *N*-oleoylethanolamine (OEA), an ethanolamine derivative of oleic acid used as an inactive lipid control. Like developmental neurite outgrowth [10], synaptamide-induced regrowth of injured axons was also GPR110- and cAMP/PKA-dependent (Figure 2E,F). Pretreatment with an *N*-terminal targeting GPR110 antibody (0.4 mg/mL) that was shown to interfere with ligand binding [10] blocked the axon regeneration. Likewise, synaptamide showed no effects on axon regrowth in cortical neurons from *gpr110 KO* mice. Inhibitors of cAMP production (SQ22536, 10 μM) or protein kinase A (H89, 10 μM) abolished the synaptamide-stimulated axon regrowth. We observed significantly elevated *gpr110* expression within one hour after axotomy in cortical neurons (Figure 2G), indicating that *gpr110* is a rapid injury-response gene.

### 2.3. Retinal Axon Outgrowth Is Stimulated by Synaptamide through GPR110/cAMP/PKA Signaling

A mouse model of nerve injury caused by optic nerve crush (ONC) is often used as an experimental disease model for traumatic optic neuropathy or glaucoma [13]. Prior to in vivo examination of the optic nerve, we evaluated whether synaptamide can stimulate the growth of retinal ganglion cell (RGC) axons that form the optic nerve. Retinal explant cultures prepared from P0 WT or *gpr110 KO* mice were treated with 10 nM synaptamide on DIV3 in the presence or absence of inhibitors or GPR110 blocking antibody and maintained for additional 7 days. Synaptamide increased neurite growth by 1.7-fold compared to the vehicle control (Figure 3A,B). Pretreatment of the culture with *N*-terminal targeting GPR110 antibody abolished the synaptamide effect, while the control antibody (IgG) did not affect the synaptamide-induced neurite outgrowth. Moreover, synaptamide treatment did not stimulate neurite outgrowth in retinal explants from *gpr110 KO* mice. Although statistical significance was not reached, the basal neurite length appeared less in *gpr110 KO* RGC cultures compared to the WT control (Figure 3B), suggesting a role of GPR110 activation by endogenous synaptamide in promoting RGC neurite outgrowth. The observed bioactivity of synaptamide was abrogated by SQ22536 and H89, confirming essential involvement of cAMP/PKA signaling, a downstream of GPR110 activation.

### 2.4. Optic Nerve Injury Upregulates gpr110 Expression

The RGC axon growth stimulated by synaptamide in a GPR110/cAMP/PKA-dependent manner suggested that optic nerve injury may be a suitable model to test synaptamide effects on axon regeneration in adult animals. *gpr110* was expressed at a low level in the RGCs of an adult mouse retina, as revealed by in situ hybridization (Figure 4A,B). As observed in the in vitro injury model using developing cortical neurons (Figure 2), the *gpr110* expression was significantly upregulated in a 4-month-old C57BL/6N mouse retina after ONC. Within 30 min after injury, *gpr110* expression increased to an extent comparable to the level present in developing brains (P0) [10], and this elevation was sustained for the entire testing period up to 48 h after injury (Figure 4C). The induction of *gpr110* in the ganglion cell layer (GCL) after injury was clearly indicated by in situ hybridization performed at 1 h after ONC (Figure 4D,E), verifying that *gpr110* is a rapid injury-response gene.

### 2.5. Synaptamide GPR110-Dependently Induces Axon Extension after Injury In Vivo

To select the desirable dose, 1.25–2.5 mg/kg of synaptamide was injected intravitreally immediately following ONC performed on 4-month-old C57BL/6N mice. Based on the dose-dependent extension of GAP43-positive regenerating axons observed at 4 weeks after injury (Figure 5A,B), we selected 2.5 mg/kg to test the synaptamide effect in vivo. A time-dependent increase in CREB phosphorylation was observed within 1 h after synaptamide injection in WT, but not in the *gpr110 KO* retina (Figure 5C), indicating that synaptamide triggered downstream cAMP/PKA/CREB signaling in vivo through GPR110 activation, as observed with developmental cortical or RGC neurons in culture (Figure 2 and Figure 3). The effect of synaptamide injection on axon growth after ONC was visualized by labeling with Alexa 555-conjugated cholera toxin B subunit (CTB) at 4 weeks after injury (Figure 5D). Synaptamide injection stimulated axon growth after injury, as indicated by the increased population of axons with longer extension from the legion site compared to the vehicle-injected control. The increase was observed only in WT but not in *gpr110 KO* mice, indicating that synaptamide-induced axon growth after injury in vivo is also GPR110-dependent (Figure 5E).

### 2.6. Biological Effect of GPR110 Activation Is Potentiated by Chemical Modification of Synaptamide to a Stable Analogue

To improve the stability of the endogenous ligand synaptamide, we generated (4Z,7Z,10Z,13Z,16Z,19Z)-N-(2-hydroxy-2-methylpropyl)docosa-4,7,10,13,16,19-hexaenamide (A8, NCGC00248435), a methylated analogue that is resistant to hydrolysis by fatty acid amide hydrolase (FAAH) (Figure 6A,B). This analogue exhibited GPR110-dependent improvement of efficacy and potency in cAMP production compared to synaptamide (Figure 6C). The 3D-model of the GPR110 GAIN domain previously established for synaptamide binding [11] revealed that the two methyl groups of A8 introduced to the synaptamide structure fit well in the hydrophobic packet created by two proline residues, P417 and P476 (Figure 6D). This hydrophobic interaction can stabilize the existing hydrogen bonding between the polar moieties in GPR110 and synaptamide, increasing binding affinity. The binding affinity evaluated using the GBVI/WSA score [14] was better for A8 (−6.26 kcal/mol) compared to synaptamide (−5.89 kcal/mol). This synthetic ligand promoted neurite outgrowth in primary neurons (vehicle: 78 ± 20, A8 (1): 122 ± 33, A8 (5): 144 ± 40, synaptamide: 142 ± 34 µm) (Figure 7A,B) and retinal explant cultures better than synaptamide (vehicle: 107 ± 11, A8 (1): 209 ± 30, A8 (5): 302 ± 40, synaptamide: 242 ± 41 µm) (Figure 7C,D). This compound also induced in vivo optic nerve axon growth after ONC (vehicle: 97 ± 9, A8: 538 ± 16, A8-KO: 54 ± 6 µm) at an intravitreal dose as low as 0.03 mg/kg in a GPR110-dependent manner (Figure 7E,F).

## 3. Discussion

In this study we demonstrated that GPR110 ligands stimulate axon growth in axotomized cortical neurons and retinal explants in culture and after optic nerve injury in adult mice. The axon growth after injury was facilitated through the rapid induction of *gpr110* and thus the upregulating of ligand-activated GPR110/cAMP/PKA signaling, even in adult neurons.

Developing neurons can extend their axons after injury, but such regenerative ability is lost with maturity [15]. A contributing factor to this regeneration failure is the loss of rapid axon growth capacity [16], which is under developmental control [17]. GPR110, the target receptor of synaptamide, mediates the neurogenic, neuritogenic, and synaptogenic activity in developing neurons through the activation of cAMP/PKA signaling [18]. Like numerous genes that undergo dynamic changes in their expression during development [19], *gpr110* is highly expressed in neural stem cells and prenatal brains, but diminishes after birth [10], suggesting that the expression of *gpr110* in the brain is also under developmental control. Therefore, the upregulation of GPR110 signaling, a developmental mechanism of stimulated neurite growth, may be a viable strategy for promoting axonal growth after nerve injury in adulthood.

Injury is known to trigger a cascade of interrelating cellular and molecular events and affects the expression of numerous genes for not only degeneration but also reorganization and repair processes [20,21,22]. In the RGC layer of an adult retina, *gpr110* expression is low, but is significantly elevated within 30 min after optic nerve injury (Figure 4C–E). In developing cortical neurons, rapid increase in *gpr110* expression was also apparent after axotomy (Figure 2G). The rapid injury-response nature of this developmental gene for stimulated neurite growth may be an intrinsic regenerative mechanism for facilitating the repair of injured axons. Indeed, activating GPR110 with its endogenous ligand stimulated axon growth after injury both in vitro (Figure 2) and in vivo, as shown for the axotomized cortical neuron culture (Figure 2) and crushed optic nerves (Figure 5).

The neurite extension observed upon synaptamide application to the axonal compartment after axotomy (Figure 2C,D) indicates the stimulated growth of injured axons in developing neurons. It has been shown that neuronal regenerative capacity can be regulated by cAMP/PKA signaling [23]. The endogenous cAMP level is much higher in neonatal neurons compared to mature neurons, and this change in cAMP level can effectively switch the glia-derived signal from stimulatory to inhibitory for axon growth [23]. When damaged CNS axons are first primed with neurotrophins before encountering inhibitory substrates derived from CNS glia around the injury site, axonal regeneration is not inhibited and the regeneration is cAMP- and PKA-dependent [24]. It is possible that upregulating cAMP/PKA signaling by specifically targeting GPR110 at an early stage after injury allows the injured axons to overcome inhibitory signals and to extend beyond the injury site in the adult CNS.

The synaptamide level in most cases correlates with the tissue level of its precursor, DHA. In the rodent brain, the DHA content is at 14–16% of the total fatty acids, while the synaptamide level is in the range of 6–30 pmol/g tissue [18]. The human optic nerve contains DHA at about 1.5% of total fatty acids [25], suggesting that the synaptamide level in the optic nerve would be substantially lower than that observed in the brain. Although the endogenous synaptamide level in the optic nerve is unknwon, it is likely insufficient to properly activate GPR110, despite the receptor upregulation after injury. Exogenously administering GPR110 ligands may have effectively activated GPR110 and downstream cAMP signaling, allowing injured axons to overcome inhibitory signals for regeneration.

It is well-established that ONC produces mild axonal injury without affecting ocular blood flow, and the survival of greater than 40% of RGCs can be achieved, depending on the severity of crush [26]. It has been also reported that axon degeneration does not require RGC somal loss. It is possible that the activation of GPR110/cAMP signaling stimulates the sprouting of new neurites from the spared soma of traumatized neurons in addition to the regenerative capacity of injured axons. The rapid induction of GPR110 in RGCs after injury may be an intrinsic mechanism for enabling stimulated axon growth through GPR110/cAMP signaling.

The effectiveness of synaptamide observed with a single treatment after ONC suggests that synaptamide action in an early stage of injury is important in stimulating axon growth and/or preserving axon regeneration capacity. In fact, GPR110-dependent increase in CREB phosphorylation was detected 30 min after synaptamide injection (Figure 5D). Similarly, *gpr110* was quickly upregulated within 30 min after injury and remained upregulated even after 48 h (Figure 4C). A8, a FAAH-resistant stable analogue of synaptmaide (Figure 6A,B) with improved binding affinity, cAMP production, and neurite growth in vitro (Figure 6C,D, Figure 7A–D), stimulated optic nerve extension after injury at a dose nearly 100-fold less than synaptamide (Figure 7E,F). GPR 110 is a member of the adhesion GPCRs that are recognized as emerging targets for drug discovery [27,28]. Further investigation is required to verify whether the observed GPR110-dependent stimulated axon growth can result in functional recovery. If so, GPR110 ligands such as A8 may present a new therapeutic opportunity for nerve injury,

In summary, our data indicate that GPR110 activation by synaptamide or its stable analogue A8 promotes axon growth in an in vitro injury model and in an optic nerve injury model in vivo. Our finding demonstrates a new potential for remedying axon injury by employing a mechanism for developmental neurite outgrowth, namely GPR110-mediated cAMP/PKA signaling. Synaptamide or A8 activates cAMP signaling in adult neurons at the injury site where the specific target receptor GPR110 is induced, and therefore can exert its bioactivity without the considerable side effects associated with systemic intervention of cAMP metabolism. We suggest that activating GPR110 may lead to a valuable strategy for stimulating axon growth after CNS injury in adults.

## 4. Materials and Methods

### 4.1. Animals

Timed pregnant female C57BL/6 mice were obtained from the NIH-NCI animal production program or Charles River Laboratories (Germantown, MD, USA), and GPR110 (adhesion G protein-coupled receptor F1: Adgrf1) heterozygous mice of a C57BL/6 background were generated by the Knockout Mouse Project (KOMP) Repository. Animals were housed in an SPF facility and acclimated for a day before their brains were collected for the preparation of primary cortical neurons or immunohistochemistry. *gpr110 KO* mice and matching WT were generated by heterozygote mating in our animal facility. All experiments were carried out in accordance with the guiding principles for the care and use of animals approved by the National Institute on Alcohol Abuse and Alcoholism (LMS-HK13).

### 4.2. Primary Cell Culture and Neurite Length Analysis

Primary cortical neurons were prepared from C57BL/6 mouse brains according to the established protocol [10]. Briefly, cortices were isolated from P0 pups and digested with 100 U of papain for 30 min at 37 °C and mechanically disrupted by pipetting several times in neurobasal medium (Invitrogen, Carlsbad, CA, USA) supplemented with 2% B27 (Invitrogen) and a 1% glutamine/glutamax mixture (1:3) (Invitrogen). The dissociated cortical neurons were seeded in poly-D-lysine-coated 24-well plates (0.5 × 10^4^ cells/well) or 8-well glass slide chambers (2 × 10^3^ cells/well) for cAMP or neurite outgrowth analysis, respectively.

Neurons were stained for β-III tubulin (1:200, cat# 5568, Cell signaling, Danvers, MA, USA) and images were acquired using a Zeiss LSM 700 confocal laser-scanning microscope. The neurite length was measured using NIH Image J software (National Institutes of Health, Bethesda, MD, USA) as we described earlier [10].

### 4.3. Cortical Neuron Culture in Microfluidic Compartment Chamber and In Vitro Axotomy

To evaluate the somal and axonal contribution to axon growth in intact or axotomized neurons, we used a microfluidic device with two-compartment chambers separated by 500 μm, as described previously [29]. The chamber (kindly provided by Dr. In-Hong Yang at the Singapore Institute for Neurotechnology at the National University of Singapore) was uniformly coated with a mixture of polylysine (100 μg/mL) and laminin (10 μg/mL). Primary cortical neurons were loaded onto the cell body side of the device and treated with agents on DIV3 for an additional 4 days. The axonal or somal compartment was added with ligands with or without pretreatment with SQ22536 (10 μM, Sigma-Aldrich, St. Louis, MO, USA), H89 (10 μM, Sigma, CA, USA), or anti-GPR110 (0.4 mg/mL, Abmart, Berkley Heights, NJ, USA) at 60 min prior to the ligand addition. Compounds were introduced into the axonal chamber very slowly and carefully using a micropipette.

For in vitro axotomy, mouse primary cortical neurons were cultured for at least 7 days in the microfluidic chamber to allow the axons to grow sufficiently into the axonal compartment. Subsequently, axotomy was carried out by carefully applying suction with fine tip glass pipette positioned at the entrance of the axonal channel. Suction was maintained for approximately 3 s, or until all visible liquid in the axonal compartment was removed. The axonal compartment was then washed with media by placing the tip at the channel entrance and vigorously pipetting up and down to remove all detached axons and debris. Following this step, fresh medium was added to the compartment for washing and the microfluidic chamber was inspected under the microscope to ensure axotomy. Axotomized and non-axotomized control cultures were treated with agents and incubated for an additional 7 days for immunocytochemistry.

### 4.4. Quantitation of Axon Growth In Vitro

Images were acquired using a Zeiss LSM 700 confocal microscope (Zeiss, White Plains, NY, USA) and analyzed using Image J-neurite tracker software 1.34. Neurons in the microfluidic chamber were fixed by 4% paraformaldehyde and incubated with anti-β-III tubulin (1:200, Cell signaling) overnight followed by Alexa 488- or 555-labeled secondary antibody. Nuclei were counter-stained by DAPI (1:10,000, Thermo Fisher Scientific Inc., Waltham, MA, USA). Axons were recognized by β-III tubulin-positive fibers and quantified with NIH Image J-neurite tracker software 1.34 to show total axon length in a field. To rule out possible dendrite contamination in the microgrooves, we selected the cross-line between the ends of microgrooves and the axonal compartment as the reference line through the entire axonal compartment. For each experiment, at least three randomly selected axon fields were evaluated to represent each individual sample for quantification in a manner blinded to each culture condition.

### 4.5. Retinal Explant Cultures and Axon Length Measurements

Retinal explant cultures were performed according to the procedure previously described [30]. P0 mouse retina was dissected, and 3 explants were taken from each animal along the dorsal–ventral axis in the center of the nasotemporal axis. The dissected explants of retina tissues were placed onto poly-D-lysine/laminin coated glass cover slips that were treated with agents including vehicle (DMSO), synaptamide, or A8 on DIV 3. After 7 days, explants were fixed with 4% paraformaldehyde (PFA) and subsequently immunostained with antibody against β-III tubulin. Axon outgrowth was quantified using NIH Image J neurite tracker software 1.34 and the total outgrowth for each explant was normalized to the explant size. The final data are presented as the axon growth relative to control.

### 4.6. Synaptamide and A8 Application

Synaptamide and A8 were added according to our protocol as previously described [10]. Primary cortical neuron or retina explant tissues were treated with synaptamide or A8 dissolved in DMSO in the presence of 40 μM vitamin E and 0.01% BSA (Sigma). As a control treatment, DMSO in medium containing BSA and vitamin E was used. In some cases, OEA in DMSO was also used as a lipid control. Anti-GPR110 (0.4 mg/mL, Santa Cruz, CA, USA), SQ22536 (Sigma), or H89 (Sigma) were added to the culture medium 60 min prior to the addition of synaptamide.

### 4.7. Optic Nerve Crush (ONC)

Mice were anesthetized by injection of ketamine (100 mg/kg, i.p.) and xylazine (10 mg/kg, i.p.). Under a binocular operating scope, a small incision was made with spring scissors (cat. #RS-5619; Roboz, Gaithersburg, MD, USA) in the conjunctiva, beginning inferior to the globe and around the eye temporally. Caution was taken, as making this cut too deep can result in cutting into the underlying musculature (inferior oblique, inferior rectus muscles inferiorly, or the lateral rectus temporally) or the supplying vasculature. With micro-forceps (Dumont #5/45 forceps, cat. #RS-5005; Roboz), the edge of the conjunctiva next to the globe was grasped and retracted, rotating the globe nasally. This exposed the posterior aspect of the globe, allowing for the visualization of the optic nerve. The exposed optic nerve was grasped approximately 1–3 mm from the globe with Dumont #N7 cross-action forceps (cat. #RS-5027; Roboz) for 3 s to apply pressure on the nerve by the self-clamping action. The Dumont cross-action forceps were chosen because its spring action applied a constant and consistent force to the optic nerve. During the 5 s clamping, we were able to observe mydriasis. After 5 s the pressure on the optic nerve was released and the forceps removed, allowing the eye to rotate back into place.

### 4.8. Intravitreal Administration of Synaptamide or A8

Synaptamide (2.5 mg/kg), A8 (0.03 mg/kg), or vehicle (DMSO) was intravitreally injected immediately after ONC while the mice were still under anesthesia. The experimenter was blinded to the identity of the compounds, including the vehicle. Before the injection of the compounds, PBS was applied to the cornea for cleaning purposes. A pulled glass micropipette attached to a 10 µL Hamilton syringe was used to deliver 2 µL of a solution into the vitreous chamber of the eye, posterior to the limbus. Care was taken to prevent damage to the lens. The pipette was held in place for 3 s after injection and slowly withdrawn from the eye to prevent reflux. Injections were performed under a surgical microscope to visualize pipette entry into the vitreous chamber and to confirm the delivery of the injected solution.

### 4.9. Anterograde Labeling

On the third day prior to euthanasia, the animals were injected with CTB–Alexa Fluor 555-conjugated cholera toxin subunit B (CTB, Life Technologies, Carlsbad, CA, USA) as an anterograde tracer to visualize axons in the optic nerve originating from RGCs.

### 4.10. Evaluation of Axon Growth after ONC

The micrographic images of the longitudinal optic nerve sections were obtained using a Zeiss LSM700 confocal microscope under 40Xagnification with automatic stitching and analyzed using Image J in a blinded fashion. Three longitudinal Z-stack sections were analyzed per each optic nerve tissue section and each experimental group included 4–6 mice. In some cases, the optic nerve micrographs were obtained after manually stitching the image sections using Photoshop. To analyze the extent of the axon growth after optic nerve injury, the number of the CTB-labeled regenerating axons (N) that passed through the measuring point at a specific distance (d) from the lesion site was estimated using the following formula as described earlier [31]: σd = *πr*^2^ × [average axon numbers per millimeter]/*t*, where *r* is half the cross-sectional width of the nerve at the counting site (d), the average number of axons per millimeter nerve width was from three nerve sections, and *t* is the section thickness (12 μm). The number of the regenerating axons extended from the lesion site was determined at a 100 μm interval. The longest axon length was determined by analyzing the micrographs of two longitudinal optic nerve sections per animal, taken under 40X magnification using a Zeiss LSM700 confocal microscope (Zeiss).

### 4.11. RNA In Situ Hybridization

Perfused tissue sections (25 μm thickness) were processed for in situ RNA detection using the RNAscope Detection Kit (Fluorescent) (Newark, CA, USA) according to the manufacturer’s instructions (Advanced Cell Diagnostics, Hayward, CA, USA). Briefly, frozen tissue samples were placed on slides and incubated with Pretreat 1 buffer for 10 min at room temperature. Slides were boiled in Pretreat 2 buffer for 15 min, followed by incubation with Pretreat 3 buffer for 30 min at 40 °C. Slides were then incubated with the relevant probes for 2 h at 40 °C, followed by successive incubation with Amp 1 to 6 reagents. Staining was visualized with fluorescent multiplex reagents (Cat# 320851, RNA scope). The in situ probes used were *gpr110* (NM_133776.2, cat# 300031), *DapB* (EF191515, cat # 310043), and *Brn3* (NM_011143.4 cat# 414671).

### 4.12. Bodipy–Synaptamide Binding to GPR110

Bodipy–synaptamide puncta staining was performed as previously described [10]. Briefly, cortical neurons were plated on 4-well glass slide chambers and treated with 100 nM bodipy–synaptamide (green) for 30 min and fixed with 4% paraformaldehyde for 10 min at room temperature. After washing, the cells were incubated with anti-clathrin H antibody (1:200, red), followed by Alexa-555-conjugated secondary antibody for 1 h at room temperature. Images were acquired on a Zeiss LSM 700 confocal laser-scanning microscope (Zeiss).

### 4.13. Immunostaining of Optic Nerve

At 4 weeks after ONC, tissues were collected after perfusion for immunostaining. Frozen sections (25 μm thickness) were prepared using a cryostat microtome (Leica) and fixed in 4% PBS-buffered polyformaldehyde solution, and permeabilized using 0.3% Triton-X 100 and 3% goat serum in PBS for immunostaining for GAP-43, a regenerating axon marker. Tissue samples were incubated with anti-GAP43 antibody (1:500, Cell Signaling) diluted in PBS containing 3% goat serum at 4 °C overnight. After washing 3 times with PBS, samples were incubated with Alexa Fluor 488 goat anti-rabbit (1:200) secondary antibody (Molecular Probes, Thermo Fisher Scientific Inc.) for 1 h at room temperature. Samples were mounted with fluoro mounting medium (Millipore, Carlsbad, CA, USA) and images were taken with a Zeiss LSM700 confocal microscope (Oberkochen, Germany).

### 4.14. Quantitative Polymerase Chain Reaction

One microgram of total RNA was reverse transcribed in the presence of 0.5 μg of Oligo(dT)15 primer (ABI, Foster, CA, USA) for 1 h at 37 °C using a reverse transcriptase kit (Cat# Qiagen, Germantown, USA) according to the manufacturer’s instructions. One microliter of first-strand cDNA was then subjected to PCR performed on ABI9700 Cycler (Foster, CA, USA) with the following conditions: 95 °C for 3 min (1 cycle), 95 °C for 30 s, 62 °C for 1 min, 72 °C for 30 s (30 cycles), and 72 °C for 10 min. RT-PCR products of the genes of interest in the wild-type and *gpr110* KO mice were grouped and analyzed simultaneously. All reactions were repeated twice with different RT reactions. Relative expression levels were determined by normalization to the glyceraldehyde 3-phosphate dehydrogenase (*gapdh*) mRNA level using 2^−^^ΔΔ^*^C^*^t^ with logarithmic transformation.

The primers used in q RT-PCR analysis were: mouse *Gpr110*: forward, 5′-CCAAGAGAAGCCAAACCTCC-3′; reverse, 5′-TTCGATAAGCCAGCAGGATG-3′, and mouse *Gapdh*: forward, 5′-ACCACAGTCCATGCCATCAC-3′: reverse, 5′-CACCACCCTGTTGCTGTAGCC-3′.

### 4.15. cAMP Assay

cAMP analysis was performed as described previously [10]. Primary cortical cells were cultured and treated on DIV3 with synaptamide or A8 for 10 min to determine cAMP levels using a cyclic AMP XP assay kit (Cell Signaling Technology, Danvers, MA, USA).

### 4.16. Western Blot Analysis

WT or *gpr110 KO* mice were perfused intracardially with phosphate-buffered saline (PBS) to clear the blood before the brains were removed. Retina and optic nerve tissues were dissected out and homogenized in ice-cold solubilization buffer (25 mM Tris pH 7.2, 150 mM NaCl, 1 mM CaCl_2_, 1 mM MgCl_2_) containing 0.5% NP-40 (Thermo Fisher Scientific, Waltham, MA, USA) and protease inhibitors (Sigma). The protein concentrations of the lysates were determined by micro BCA protein assay (Thermo Fisher Scientific). Samples for SDS-PAGE were prepared at 1 μg protein/μL concentration using 4X SDS-PAGE buffer (Thermo Fisher Scientific, Waltham, MA, USA) and 20 μg of protein was loaded onto each well. Proteins were separated by SDS-PAGE on 4–15% polyacrylamide gels (Thermo Fisher Scientific) and transferred onto a PVDF membrane (Thermo Fisher Scientific, Waltham, MA, USA). After treating with blocking buffer containing 0.01% Tween-20, 10% BSA (Thermo Fisher Scientific) for 1 h at room temperature, blots were incubated with primary antibody diluted in blocking buffer (anti-CREB, anti-phospho-CREB, rabbit anti-GAPDH 1:1000, Cell Signaling Technology) overnight at 4 °C followed by HPR-conjugated secondary antibodies (1:5000, Cell Signaling Technology) for 1 h at room temperature. The detection was performed using the KODAK Imaging system.

### 4.17. Preparation of A8 (4Z,7Z,10Z,13Z,16Z,19Z)-N-(2-Hydroxy-2-methylpropyl)docosa-4,7,10,13,16,19-hexaenamide (NCGC0024843)

100 mg (0.304 mmol) of (4Z,7Z,10Z,13Z,16Z,19Z)-docosa-4,7,10,13,16,19-hexaenoic acid (Enzo) and 32.6 mg (0.365 mmol) of 1-amino-2-methylpropan-2-ol (Matrix Scientific) were taken in a nitrogen filled dry 25 mL pear shaped flask charged with a magnetic stirrer. 2 mL dichloromethane (Sigma Aldrich) was added, followed by 127 mg (0.335 mmol) hexafluorophosphate azabenzotriazole tetramethyl uronium (HATU, Combi-Blocks) and 0.085 mL (0.609 mmol) triethylamine (Sigma Aldrich). The reaction was stirred for 10 min at room temperature and then diluted with more dichloromethane (~20 mL) washed with saturated aqueous ammonium chloride (~20 mL) and then saturated aqueous sodium bicarbonate. The dichloromethane layer was separated, dried with magnesium sulfate, filtered, and concentrated in vacuo. The residue was purified via flash silica gel chromatography (0 to 75% ethyl acetate in dichloromethane); fractions could be visualized by thin layer chromatography followed by staining with para-anisaldehyde stain or iodine. Fractions containing the product were combined, dried via rotary evaporation, and then under high vacuum to obtain 105 mg (0.263 mmol, 86% yield) of A8 which was protected from light and stored under nitrogen at −80 °C.

^1^H spectra were recorded on a Varian Inova 400 MHz spectrometer (Palo Alto, CA, USA). Chemical shifts are reported in ppm with the solvent resonance as the internal standard (MeOH-d_4_ 3.31 ppm for Me). Data are reported as follows: chemical shift, multiplicity (s = singlet, d = doublet, t = triplet, q = quartet, m = multiplet), coupling constants, and number of protons, apparent multiplicity (if observed). ^1^H NMR (400 MHz, methanol-*d*_4_) δ 5.45–5.24 (m, 12H), 3.18 (s, 2H), 2.91–2.80 (m, 10H), 2.45–2.33 (m, 2H), 2.29 (ddd, *J* = 7.9, 6.9, 1.1 Hz, 2H, app triplet), 2.09 (dqdd, *J* = 7.5, 6.7, 1.4, 0.7 Hz, 2H, app pentet), 1.16 (s, 6H), 0.97 (t, *J* = 7.5 Hz, 3H). Analytical purity analysis and retention times (RT) reported here were performed on an Agilent LC/MS (Agilent Technologies, Santa Clara, CA, USA). A Phenomenex Luna C18 column (3 µ, 3 × 75 mm) was used at a temperature of 50 °C. The solvent gradients are mentioned for each compound and consist of a percentage of acetonitrile (containing 0.025% trifluoroacetic acid) in water (containing 0.05% trifluoroacetic acid). A 4.5 min run time at a flow rate of 1 mL/min was used. A8 was at >95% purity based on the above methods and had RT = 3.9 min. Mass determination was performed using an Agilent 6130 mass spectrometer with electrospray ionization (ESI) in the positive ion mode. Calculated mass for C_26_H_42_NO_2_^+^ (M+H)^+^ was 400.3, found 400.3.

### 4.18. Docking Studies

The 3D structure of GPR110 was generated using the I-TASSER program as described previously [10,11]. Docking of synaptamide and A8 to the GAIN domain of GPR110 was performed using the MOE program [14]. The induced fit protocol was used for ligand docking, and the binding affinity was evaluated using the GBVI/WSA score [14].

### 4.19. Statistical Analysis

Data were analyzed using GraphPad Prism 7 software (San Diego, CA, USA) and all data are presented as mean ± s.e.m and are representative of at least two independent experiments. Statistical significance was determined by an unpaired Student’s *t* test or one-way ANOVA. * *p* < 0.05, ** *p* < 0.01 and *** *p* < 0.001.

## Figures and Tables

**Figure 1 ijms-22-03386-f001:**
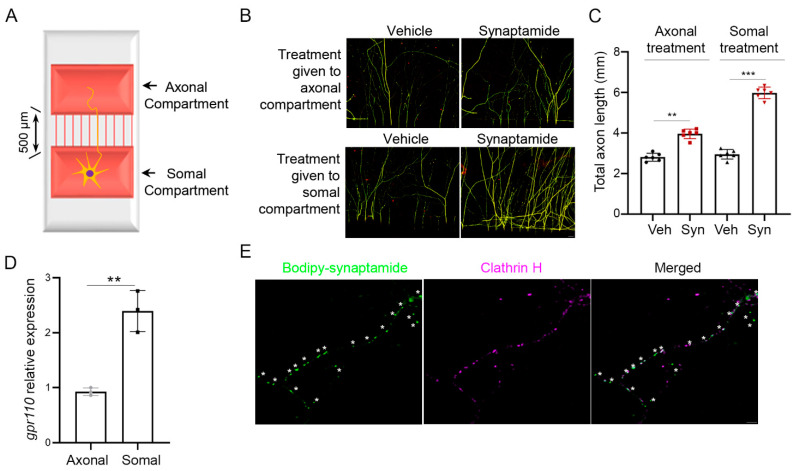
Axon growth promoted by either somal or axonal treatment of synaptamide. (**A**) Schematic representation of a microfluidic culture platform with two chambers separated by 500 μm of thin grooves that was used to culture primary cortical neurons from the mouse brain at postnatal day 0 (P0). (**B**) β-III tubulin immunofluorescence micrographs showing that axon outgrowth in primary cortical neurons was induced by the application of 10 nM synaptamide to the axonal or somal compartment on DIV3 and cultured for an additional 4 days. (**C**) Quantitative results of the total length of axons entered in the axonal compartment. Synaptamide application to the cell body or axonal compartment resulted in a 2.2- or 1.3-fold increase in axon growth, respectively, compared to the vehicle control. (**D**) Relative quantification of *gpr110* expression in soma and axon determined by RT-PCR after collecting samples separately from the somal or axonal compartment of the two-chamber culture platform. (**E**) Internalization of GPR110 visualized by a fluorescent ligand bodipy–synaptamide. After 30 min stimulation of DIV 3 cortical neurons with 20 nM bodipy-synaptamide, internalized GPR110 was detected as fluorescent endocytic puncta (green) overlapping with an endocytic marker clathrin H (magenta), confirming the axonal presence of GPR110. Asterisks (*) represent endocytic GPR110 receptor puncta. Data are expressed as mean ± s.e.m. with n = 6 for (**B**) and n = 3 for (**D**) per group. ** *p* < 0.01, *** *p* < 0.001 by t-test; Veh, Vehicle (DMSO); Syn, synaptamide. Scale bar, 20 μm (**B**), 1 μm (**E**).

**Figure 2 ijms-22-03386-f002:**
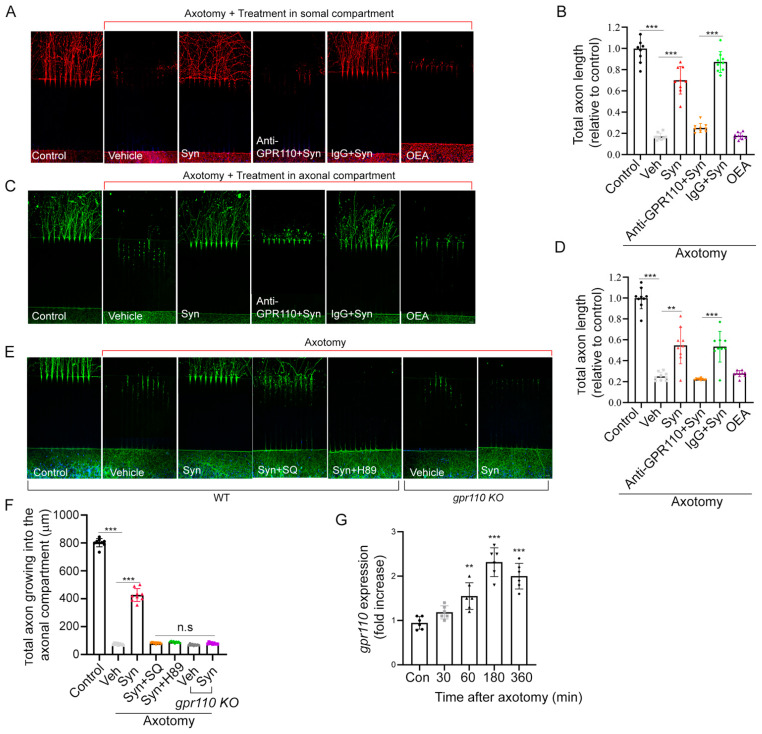
Synaptamide stimulates axon regeneration in vitro through GPR110/cAMP/PKA-dependent signaling. (**A**–**D**), Stimulated axon regrowth by somal (**A**,**B**) or axonal treatment (**C**,**D**) with 10 nM synaptamide, visualized by β-III tubulin immunostaining and quantified by measuring the total axon length in the axonal compartment 7 days after axotomy. (**E**,**F**), Effect of cAMP (SQ22536, 10 µM) and PKA (H89, 10 µM) inhibitors, *N*-terminal targeting GPR110 antibody (0.4 mg/mL), or *gpr110 KO* on synaptamide-induced axon regrowth with quantitative results normalized to the non-axotomized control. G, Time-dependence of axotomy-induced increase in *gpr110* expression measured by qRT-PCR. Data are expressed as mean ± s.e.m. with n = 9 for (**B**,**D**,**F**) and n = 6 for (**G**) per group. ** *p* < 0.01, *** *p* < 0.001, Con, non-axotomized control; Veh, vehicle (DMSO); Syn, synaptamide. Scale bars, 20 μm. n.s, statistically not significant.

**Figure 3 ijms-22-03386-f003:**
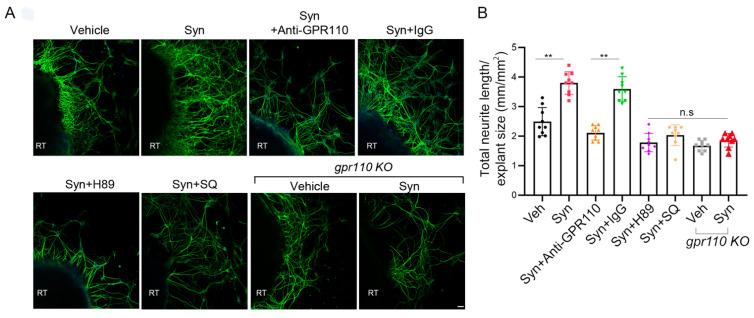
Retinal neurite outgrowth is stimulated by synaptamide through the GPR110/cAMP/PKA signaling pathway. (**A**), Neurite outgrowth from cultured P0 retinal explants increased by 10 nM synaptamide and inhibited by N-terminal targeting GPR110 antibody (0.4 mg/mL), SQ22536 (10 µM), H89 (10 µM), or *gpr110 KO*. The cultures were treated with agents on DIV 3 for 7 days and immunostained for β-III tubulin. (**B**), Total neurite length per field adjusted for the explant size. Data are expressed as mean ± s.e.m with n = 9 per group. ** *p* < 0.01 by one-way ANOVA. RT, retinal explant tissue. Scale bar, 50 μm. n.s, statistically not significant.

**Figure 4 ijms-22-03386-f004:**
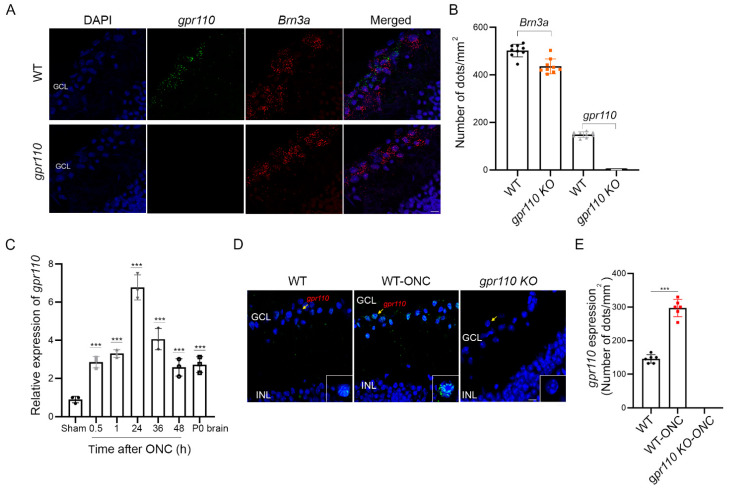
Optic nerve injury upregulates *gpr110* expression. (**A**) Expression of *gpr110* in RGCs from 4-month-old mice detected by in situ hybridization using probes for *gpr110* (green) and a RGC marker *Brn3a* (red), overlaid on DAPI-stained nuclei (blue). (**B**) Quantitation of *gpr110* expression. (**C**–**E**) Induction of *gpr110* in the retina tissue after ONC determined by qRT-PCR (**C**) and by in situ hybridization of the ganglion cell layer (GCL) with micrographic (**D**) and quantitative results (**E**). Data are expressed as mean ± s.e.m with n = 9 (**B**) n = 3 (**C**) and *n* = 6 (**E**) per group. *** *p* < 0.001 by one-way ANOVA. GCL, ganglion cell layer; INL, inner nuclear layer. Scale bars, 20 μm (**A**,**D**).

**Figure 5 ijms-22-03386-f005:**
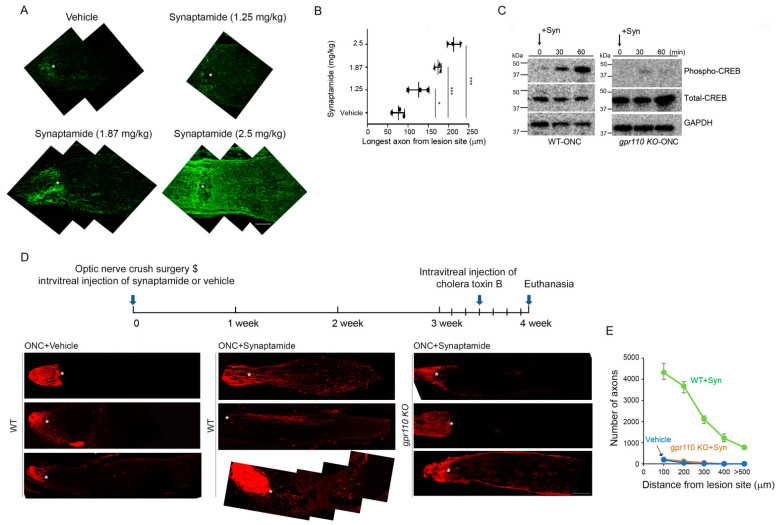
Synaptamide GPR110-dependently induces axon growth after optic nerve injury. (**A**) Dose-dependent effect of synaptamide on stimulated axon growth after injury. ONC was performed on 4-month-old BALB/C mice and synaptamide was administrated intravitreally at the doses indicated. Longitudinal sections through the optic nerve were collected at 4 weeks after ONC and regenerating axons were visualized by anti-GAP43 antibody (green, 1:500). Lesion site was marked by asterisks (*). (**B**) Quantitative measurement of axon length extended from the lesion site after synaptamide injection at the doses indicated. (**C**) Time-dependent increase in CREB phosphorylation after synaptamide injection into 4-month-old WT but not *gpr110 KO* mouse retinas following ONC. (**D**) Optic nerve tissue sections collected at 4 weeks after ONC and visualized by CTB, indicating that intravitreal injection of synaptamide (2.5 mg/kg) induced axon growth in WT but not in *gpr110 KO* mice. The optic nerve micrographs shown in A and at the bottom panel of the ONC+Synaptamide group in D were obtained after the manual stitching of image sections using Photoshop. (**E**) The number of the axons extended to a 100–600 micrometer distance from the lesion site, indicated by the asterisk. Data are expressed as mean ± s.e.m with n = 3 (**A**) and n = 4 (**E**) per group. * *p* < 0.05, *** *p* < 0.001 by one-way ANOVA. Scale bars, 100 μm (**A**,**D**).

**Figure 6 ijms-22-03386-f006:**
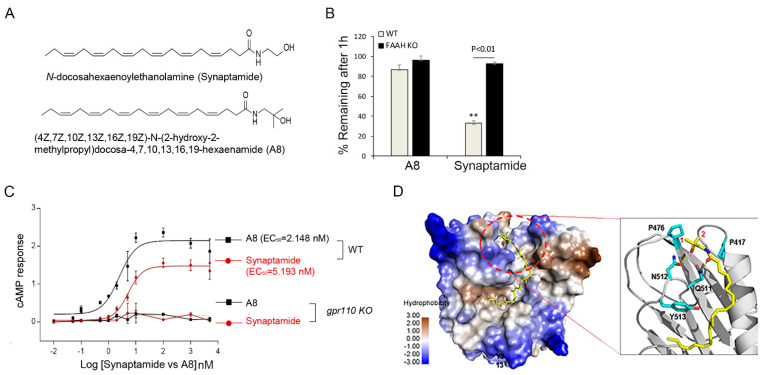
A8, a FAAH-resistant structural analogue of synaptamide, exhibited improved cAMP production and receptor binding. (**A**) Chemical structures of synaptamide and its stable analogue A8 ((4Z,7Z,10Z,13Z,16Z,19Z)-N-(2-hydroxy-2-methylpropyl)docosa-4,7,10,13,16,19-hexaenamide). (**B**) Synaptamide and A8 remaining after incubation with brain homogenates from WT or fatty acid amide hydrolase (FAAH) KO mice for 1 h, showing that A8 is resistant to FAAH hydrolysis. (**C**) Dose-dependent increase in cAMP production elicited by synaptamide and A8. Cortical neurons derived from P0 brains were stimulated for 15 min on DIV3 for cAMP measurement. (**D**) Improved binding ability of A8 to GPR110 predicted by the binding model of A8 at the GAIN domain of GPR110. The protein surface is rendered in the color of hydrophobicity. A8 is shown in sticks of yellow carbon atoms, and key residues at the hydrophobic pocket are shown in sticks of cyan carbon atoms. O and N atoms are shown in red and blue, respectively. The two methyl groups (labeled with 1 and 2) of A8 fit well in the hydrophobic pocket formed by P470 and P417. The resulting hydrophobic interaction stabilizes the hydrogen bonding between the polar moieties of GPR110 and synaptamide or A8 (dotted yellow lines). ** *p* < 0.01 vs. A8.

**Figure 7 ijms-22-03386-f007:**
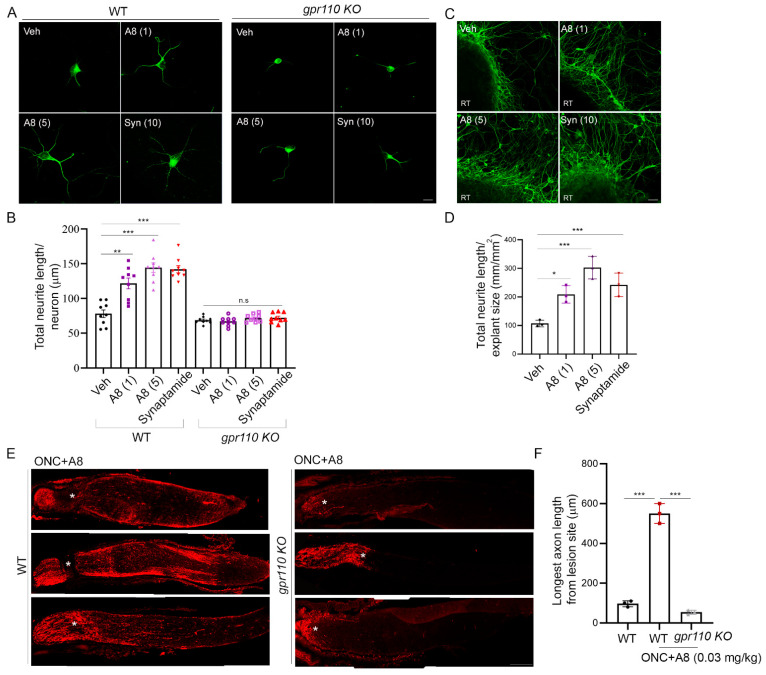
A8, a FAAH-resistant structural analogue of synaptamide, improved neurite outgrowth and axon growth after injury. (**A**,**B**) Improved neurite outgrowth by A8 in cultured cortical neurons from WT but not from *gpr110 KO* mice, indicating the GPR110-dependent nature of A8 bioactivity. Neurite length was evaluated after treating DIV0 cultures with synaptamide (10 nM) or A8 (1 or 5 nM) for 3 days. (**C**,**D**) RGC neurite growth in retinal explant cultures promoted by synaptamide or A8 as indicated by the quantification of axon length after immunostaining with anti β-III tubulin antibody. (**E**,**F**) GPR110-dependent stimulatory effect of intravitreal injection of A8 at 0.03 mg/kg on axon extension, shown by the micrographic images of CTB-labeled axons (**E**) and quantitative data showing the longest axon length (**F**) obtained at 4 weeks after ONC. Some of the optic nerve micrographs (**E**) were obtained after the manual stitching of image sections using Photoshop. Data are expressed as mean ± s.e.m with n = 9 (**A**) and n = 3 (**D**,**F**) per group. RT, retinal tissue; * *p* < 0.05, ** *p* < 0.01, *** *p* < 0.001; Scale bars, 10 μm (**A**), 50 μm (**C**) and 100 μm (**E**).

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
