# Peer review of "Ligand-Induced GPR110 Activation Facilitates Axon Growth after Injury"

_ijms, 2021, doi:10.3390/ijms22073386_

Round 1

Reviewer 1 Report

This is an interesting and very clear paper on the axon regeneration properties mediated by ligands of G-protein coupled receptor GPR110 and represents a substantial amount of work. The GPR110 ligand Synaptamide was tested in an in vitro culture system and validated in a mouse optic nerve crush model. Like the majority of these studies, the actual regrowth is modest but comparable to PTEN or PTP sigma modulation. A simple SAR modification was used to successfully produce a more stable version of Synaptamide. Modulation of cyclic nucleotides in regenerating axons has been investigated since the 1990s, when they were found to switch repulsion to attraction.  Synaptamide appears to be a useful way to modulate cyclic nucleotide levels via GPR110 and therefore has the potential to enhance axonal regeneration, likely in combination therapies. The paper appears very suitable for publication in IJMS.

Minor suggestions:

To help color blind readers, it is suggested that Fig. 1E is changed from red/green to magenta/green colors or similar.

It is unclear why red fluorescence is used in Fig. 2A and green in 2C and 2E. I assume it does not signify a difference in staining and it may distract the reader.

I wouldn’t normally say this but increasing the contrast in Fig. 4A would help interpreting this figure. The fluorescence is currently quite hard to see. The ISH signal in 4D is only apparent after significant zooming in.

The source of the beta-tubulin antibody lacks a catalog number and the name of company may be incorrectly abbreviated. I cannot locate the source of synaptamide used in the paper. Otherwise, antibody and chemical reagents are clearly identified in the methods.

Author Response

  1. To help color blind readers, it is suggested that Fig. 1E is changed from red/green to magenta/green colors or similar.

 Figure 1E has been changed as suggested.

  1. It is unclear why red fluorescence is used in Fig. 2A and green in 2C and 2E. I assume it does not signify a difference in staining and it may distract the reader.

The purpose was to distinguish the somal or axonal application of reagents after axotomy. As the reviewer assumed, it does not make any difference and we prefer to leave it as it is. 

  1. I wouldn’t normally say this but increasing the contrast in Fig. 4A would help interpreting this figure. The fluorescence is currently quite hard to see. The ISH signal in 4D is only apparent after significant zooming in. 

The images are now fixed as suggested.

  1. The source of the beta-tubulin antibody lacks a catalog number and the name of company may be incorrectly abbreviated. I cannot locate the source of synaptamide used in the paper. Otherwise, antibody and chemical reagents are clearly identified in the methods. 

The information has been added (Line 324).

Reviewer 2 Report

This research article found an up-regulation of GPR110 receptor after injury and stimulating its signalling promotes developmental axon growth and regeneration in the central nervous system. The authors identified GPR110 as receptor and cAMP/PKA as downstream pathway responsible for axon growth mediated by synaptamide and the A8 analogue peptide. The activation state of regeneration-associated receptors is an important aspect of axon regeneration.

The experimental designs are excellent. All the appropriate controls are included in the performed experiments. Furthermore, the extend of optic nerve regeneration is shown for each animal in the experimental groups; this should become the standard in the field of axon regeneration.

In my opinion, no further experiments are needed to support the conclusions of this article.

One suggestion that could further improve the merits of this article is to examine the effect of synaptamide / A8 on RGC survival in retinal wholemounts after optic nerve crush, because an optic nerve crush leads to death of RGCs. 

Below are suggestions for textual changes to improve the article:
1) The introduction is very short. Please could you expand the introduction with a few paragraphs? Is GPR110 expressed in the regenerating peripheral nervous system? Describe the developmental down regulation of GPR110 in the CNS. What are the ligands regulating the activation state of GPR110 (both activating and inhibiting?). Are these ligands differently expressed in the CNS and PNS? And after injury?
2) Comment for the discussion. The article shows an upregulation of GPR110 after axotomy, while the intrinsic regeneration capacity of RGC is relatively low. Please could you comment on the endogenous expression levels of synaptamide in the injured optic nerve? Are there perhaps molecules that supress signalling of the injury-induced GPR110 that prevent endogenous axon regeneration by RGC?
3) Please clarify the number of cells examined in figure 7B. Primary cortical neurons are a diverse population of neurons and hence a relative large number of cells are needed in order to reliably establish the total neurite length per cell.
4) Please mention the length of axon growth and regeneration in the text of the results section.
5) The puncta in panel 1E and figure 4 are difficult to see. The neurites in panel 7A are also poorly visible. Please could you increase the colour intensity or convert those single colour images in grey scale?
6) I would like to notify you that the Greek symbols in β3-tubulin, μm, and potentially others, got lost during the upload of manuscript to the journal.

I advice the editor to accept this manuscript for publication with minor text editing. 

Author Response

  1. The introduction is very short. Please could you expand the introduction with a few paragraphs? Is GPR110 expressed in the regenerating peripheral nervous system? Describe the developmental down regulation of GPR110 in the CNS. What are the ligands regulating the activation state of GPR110 (both activating and inhibiting?). Are these ligands differently expressed in the CNS and PNS? And after injury?

The information regarding the developmental expression of GPR110 is already indicated in the discussion (lines 239-240). Since it is a newly deorphanized GPCR, ligands that can regulate the GPR110 signaling are yet to be discovered.  Further studies are also required to elucidate the relative expression of GPR110 in the central and peripheral nervous system as well as the induction capacity after injury.

We have added the following sentence in the introduction (Lines 36-37).

“GPR110 is an adhesion GPCR, and its expression in the brain is highest during development but diminishes in adulthood [10].”

  1. Comment for the discussion. The article shows an upregulation of GPR110 after axotomy, while the intrinsic regeneration capacity of RGC is relatively low. Please could you comment on the endogenous expression levels of synaptamide in the injured optic nerve? 

The endogenous synaptamide level in the optic nerve has not been determined.  In most cases, the synaptamide level responds to the tissue level of its precursor DHA, and the level of synaptamide is in the range of 6 - 30 pmol/g tissue in the rodent brain (Kim and Spector) where the DHA content is found in the range of 14-16%. The human optic nerve contains DHA at 1.5 % of total fatty acids (Acar et al. 2012), suggesting that the endogenous synaptamide level in the optic nerve would be substantially lower than that observed in the brain. The endogenous synaptamide in the optic nerve might not be sufficient to properly activate GPR110 despite the upregulation of the receptor after injury.  It is likely that GPR110 and downstream cAMP signaling is activated effectively by exogenous GPR110 ligands, allowing injured axon to overcome inhibitory environment and to regenerate.

Are there perhaps molecules that suppress signaling of the injury-induced GPR110 that prevent endogenous axon regeneration by RGC?

It is well-documented that axon regeneration is hampered by myelin-associated inhibitory environment as indicated in the introduction and discussion. Again, any interaction with GPR110 signaling is presently unknown.  It is possible that the endogenous level of synaptamide is not sufficient to enable axon regeneration despite the upregulation of GPR110 expression. 

We have included the following paragraph in the discussion (Lines 264-272).

“The synaptamide level in most cases correlates with the tissue level of its precursor DHA. In the rodent brain, the DHA content is at 14 -16% of the total fatty acids while the synaptamide level is in the range of 6 - 30 pmol/g tissue [18]. The human optic nerve contains DHA at about 1.5 % of total fatty acids [25], suggesting that the synaptamide level in the optic nerve would be substantially lower than that observed in the brain. Although the endogenous synaptamide level in the optic nerve is unknwon, it is likely insufficient to properly activate GPR110 despite the receptor upregulation after injury. Exogenously administering GPR110 ligands may have effectively activated GPR110 and downstream cAMP signaling, allowing injured axons to overcome inhibitory signals for regeneration.”

  1. Please clarify the number of cells examined in figure 7B. Primary cortical neurons are a diverse population of neurons and hence a relative large number of cells are needed in order to reliably establish the total neurite length per cell. 

Figure 7B has the n number of 9 as indicated (Line 222).

  1. Please mention the length of axon growth and regeneration in the text of the results section. The axon length is now indicated in the result section (Lines 193-197).
  2. The puncta in panel 1E and figure 4 are difficult to see. The neurites in panel 7A are also poorly visible. Please could you increase the colour intensity or convert those single colour images in grey scale?

We adjusted the brightness and contrast to improve the images as suggested.

6) I would like to notify you that the Greek symbols in β3-tubulin, μm, and potentially others, got lost during the upload of manuscript to the journal.

The mislabeled symbols have been corrected. Thank you.